# Effect of Information Disclosure Policy on Control of Infectious Disease: MERS-CoV Outbreak in South Korea

**DOI:** 10.3390/ijerph17010305

**Published:** 2020-01-01

**Authors:** Jin-Won Noh, Ki-Bong Yoo, Young Dae Kwon, Jin Hyuk Hong, Yejin Lee, Kisoo Park

**Affiliations:** 1Department of Health Administration, College of Health Science, Dankook University, Cheonan 31116, Korea; jinwon.noh@gmail.com; 2Department of Health Sciences, Global Health, University Medical Centre Groningen/University of Groningen, 9713 GZ Groningen, The Netherlands; 3Department of Health Administration, College of Health Sciences, Yonsei University, Wonju 26493, Korea; ykbong@yonsei.ac.kr; 4Department of Humanities and Social Medicine, College of Medicine and Catholic Institute for Healthcare Management, the Catholic University of Korea, Seoul 06591, Korea; healthcare@catholic.ac.kr; 5Department of Biostatistics, Korea University College of Medicine, Seoul 02841, Korea; hongjh0513@korea.ac.kr; 6Department of Healthcare Management, Eulji University, Seongnam 13135, Korea; yiye1110@gmail.com; 7Institute for Occupational & Environmental Health, Korea University College of Medicine, Seoul 02841, Korea

**Keywords:** Middle East respiratory syndrome, MERS-CoV, infectious disease, prevention, information disclosure policy

## Abstract

This study examined the effect of disclosing a list of hospitals with Middle East respiratory syndrome coronavirus (MERS-CoV) patients on the number of laboratory-confirmed MERS-CoV cases in South Korea. MERS-CoV data from 20 May 2015 to 5 July 2015 were from the Korean Ministry of Health & Welfare website and analyzed using segmented linear autoregressive error models for interrupted time series. This study showed that the number of laboratory-confirmed cases was increased by 9.632 on 5 June (*p* < 0.001). However, this number was significantly decreased following disclosure of a list of hospitals with MERS-CoV cases (Estimate = −0.699; *p* < 0.001). Disclosing the list of hospitals exposed to MERS-CoV was critical to the prevention of further infection. It reduced the number of confirmed MERS-CoV cases. Thus, providing accurate and timely information is a key to critical care response.

## 1. Introduction

The first case of Middle East respiratory syndrome coronavirus (MERS-CoV) was reported in Saudi Arabia in 2012 [1]. Since then, the World Health Organization has been notified of 1368 cases of MERS-CoV infection, including at least 490 cases that led to death [2,3]. The first case of MERS-CoV infection in a 68-year-old man who had a history of travel to the Middle East was confirmed in South Korea on 20 May 2015 [4]. As of July 2015, 186 cases of MERS-CoV and 36 related deaths (fatality rate 19.4%) have been reported in South Korea [5]. This fatality rate is the second highest in the world.

MERS-CoV crisis not only exerted massive social and economic impact on South Korea, but also revealed drawbacks in South Korean public health and infection control of the healthcare system. Large-scale MERS-CoV infection in South Korea occurred by intra- and inter-hospital transmission [6]. The first MERS-CoV patient was exposed to other patients, healthcare providers, and visitors during his first hospital visit due to inadequate management of patients with infectious diseases in South Korea [5]. Clearly, central and municipal governments failed to deal with the current MERS-CoV epidemic. They also failed to control private hospitals. Consequently, the basic cause of the spread of MERS-CoV by infected patients as they moved throughout the country was associated with governments’ initial countermeasures.

The Ministry of Health and Welfare supervises infectious-disease disaster management in South Korea while the Ministry of Public Safety and Security supports disaster control [6]. After the Ebola crisis in 2014, the government installed and operated airborne-infection isolation rooms at local district public hospitals to treat patients with acute infectious disease [7]. However, incompleteness has been exposed despite such measure [6]. In addition, airborne infection is identified as a serious problem in South Korean MERS-CoV guidelines [6]. The definition of “close contact” with respect to MERS-CoV response strategies used by the Korea Centers for Disease Control and Prevention (KCDC) was unclear and incompatible with that of the World Health Organization. Such incorrect definition resulted in serious mistakes made by the KCDC when making contact tracing and management in the early epidemic stage of MERS-CoV [6,8].

The government revised its policy to disclose information in accordance with the public’s opinion that nondisclosure of information would in fact cause MERS-CoV to spread and generate public anxiety. However, the South Korean government restricted and controlled information regarding MERS-CoV during the initial response. It has been claimed that inadequate information disclosure was the main reason for the spread of MERS-CoV. The lack of an appropriate management system and a prevention strategy for infectious disease has been raised as additional reason for the initial failure to contain the outbreak [6].

The purpose of this study was to evaluate the effect of policy to avoid disclosure of information, the most controversial issue regarding the initial response, on the spread of MERS-CoV in South Korea. The study sought to determine whether changing this strategy and releasing a list of hospitals treating infected individuals was associated with containment of the disease [9].

## 2. Methods

Data were obtained from the MERS-CoV information website (http://mers.go.kr/) managed by the Korean Ministry of Health & Welfare. This website provides information regarding the statistics and management status for MERS-CoV [10]. These data contained daily statistics regarding numbers of laboratory-confirmed cases, quarantined individuals, patients undergoing treatment, and deaths and other countries’ MERS-CoV related information. We included daily numbers of laboratory-confirmed cases and quarantined individuals from 20 May 2015 to 5 July 2015 in a time-series analysis. 20 May was chosen because it was the day on which MERS-CoV was confirmed in the first patient in Korea. There were no additional laboratory-confirmed cases after 5 July 2015. This study was approved by the Institutional Review Board of the institution with which the first and second authors were affiliated.

Daily number of laboratory-confirmed cases released by the Korean government was included as a dependent variable. The date of disclosure of the hospital list and daily number of quarantined individuals were included as independent variables. The daily number of quarantined individuals was used to adjust for the epidemic status of MERS-CoV in Korea. Individuals who had been exposed to the virus or in contact with either MERS-CoV patients or those with suspected MERS-CoV were defined as quarantined individuals. They were required to submit specimens for MERS-CoV diagnosis. Such individuals were isolated for 14 days in their homes or a medical facility depending on their symptoms. This study analyzed MERS-CoV data from 20 May 2015 to 5 July 2015.

Segmented linear autoregressive error models for interrupted time series were used to assess the effect of disclosure of hospital names and disease-related hospital management on the number of laboratory-confirmed MERS-CoV cases [11]. Segmented regression analysis is a useful statistical method for evaluating the longitudinal effect of policy intervention in quasi-experimental designs without a control group [12]. SAS 9.4 PROC AUTOREG was used to create segmented linear autoregressive error models via SAS Version 9.4 (SAS Institute, Inc., Cary, NC, USA). The order of the autoregressive error models was selected using stepwise auto regression with BACKSTEP option in SAS 9.4. All parameters were estimated via maximum likelihood estimators. Daily numbers of quarantined individuals and laboratory-confirmed MERS-CoV cases were used as units of analysis. The equation for laboratory-confirmed MERS-CoV cases is shown below (Equation (1)). The stepwise autoregressive process selected a seventh-order subset model with nonzero parameters at lag 7, AR [7] error.
(1)$$\begin{array}{cc} ConfirmedCases_{t} & =\beta_{0}+\beta_{1}Time_{t}+\beta_{2}Disclosure_{t}+\beta_{3}Time\_after\_disclosure_{t}\\ & +\beta_{4}Quarantines_{t-8}+e_{t} \end{array}$$
where *t* was time period (day), *Time* was a continuous variable initiated on 20 May 2015 (daily), *disclosure* was binary variable representing disclosure of the hospital list (0 before 5 June; 1 after 5 June 2015), *Quarantines* was a continuous variable representing the daily number of quarantined individuals, *Time_after_disclosure* was a continuous variable initiated on 5 June 2015 (daily), and $e_{t}$ was an error term.

## 3. Results

Figure 1 shows daily statistics from 20 May 2015 to 5 July 2015. The first MERS-CoV patient and secondary case were confirmed on 20 May 2015. Twenty-three cases of MERS-CoV were confirmed on 7 June 2015. This was the highest number of confirmed cases recorded on a single day during the study period. The highest number of quarantined patients was recorded on 17 June 2015. Thereafter, it showed a downward trend. The highest number of deaths was recorded on both 12 June 2015 and 17 June 2015. No additional cases were confirmed after 4 July 2015.

Results of segmented regression for confirmed cases are shown in Table 1. Baseline time trend showed a significant increase (Estimate = 0.316; *p* < 0.001). The number of confirmed cases was increased by 9.632 on 5 June (*p* < 0.001). It showed a significant downward trend following disclosure of hospital list (Estimate = −0.699; *p* < 0.001). The spread of the disease ceased because potential MERS-CoV patients were screened. Figure 2 shows results and predictions of the statistical analysis. Results of sensitivity analyses are shown in Table 2. They were almost consistent with those shown in Table 1.

## 4. Discussion

The study aimed to analyze the effect of disclosing a list of hospitals exposed to MERS-CoV on the spread of the disease. On the day when the list of hospitals that MERS-CoV patients had visited or attended for treatment was disclosed, the number of confirmed cases was increased. However, a downward trend was observed thereafter. This result implies that disclosing the hospital list was very effective in preventing people from further infection of MERS through timely provision of information.

The following two mechanisms were presumably involved in the downward trend of infection. First, the information dissemination was likely to lower the possibility of the public’s further contracting MERS-CoV, as it helped members of the public to avoid exposed hospitals with the given information by the health authority [13]. Following a news article, hospital utilization had decreased by up to 50% since 20 May 2015. This might have occurred because citizens were reluctant to visit hospitals as they were afraid of contracting the virus [2]. Therefore, the way citizens had stayed vigilant not to catch the virus was likely to be strengthened after the disclosure of the hospital names. Second, since the citizens had received the information regarding the affected hospitals, people who had visited one of those hospitals or contact friends and family who went to the affected medical facilities would be more likely to report to the KCDC when they experienced MERS symptoms such as fever, vomit, sore throat, vomiting, and diarrhea and follow the prevention and control guidance voluntarily and effectively [14]. These mechanisms are supposed to have contributed to the reduction in the number of confirmed cases.

This study showed that the Korean health authorities’ decision to disclose information regarding hospitals exposed to MERS-CoV contributed to the prevention of further MERS-CoV infection during the outbreak, despite the criticism that the disclosure occurred too late. In addition, findings of this study indicate that health authorities should disseminate information regarding facilities affected by an outbreak of infectious diseases as soon as possible to ensure that people are able to avoid the risk of infection. This confirms the risk communication principle of timely provision of information in coping with outbreaks is so important. It is inappropriate to withhold information that is crucial to decision making for members of the public not only from a communication perspective, but also from epidemiological and clinical perspectives. Therefore, authorities should pay greater attention to risk communication [15]. Following the outbreak of SARS in Singapore, most people emphasized the importance of current study results and the disclosure of crucial information to the public and the media, as to what would have helped during the crises. This should be documented legally as soon as possible to ensure that all disease-related information is provided to the public and the media automatically and systemically when outbreaks occur. It should be noted that on January 6 2016, the newly implemented Infection Disease Prevention Act provided the KCDC with the power to release lists of medical facilities affected by infectious diseases automatically when outbreaks occur.

These results also suggest that Korea health authorities should consult with experts and produce a crisis and risk communication guide to develop communication strategies such as informing the public and providing training for communicators who possess the necessary competence and expertise [6]. Overall, the provision of accurate and prompt information is one of the most important factors in saving lives and protecting people during public health emergencies. Naylor et al. have suggested that communication between authorities and the public is the cornerstone of crisis management in the healthcare and public health system [13].

This study had several limitations. First, we did not analyze the effects of other noteworthy variables such as the increased number of health authorities’ staff, medical facilities’ capacity bolstering, the strengthened engagement of media, and other private sectors over time. For example, increases in the number of staff members in the Central MERS Management Task Force played an important role in preventing and controlling the outbreak. Therefore, the size of the MERS Rapid Response Team dispatched to exposed hospitals should be included in further studies. Besides, hospitals regardless of having MERS patients at that time in Korea had been more involved in detecting and controlling outpatients and visitors as time went by. Therefore, this factor could be one of the reasons to help decrease the number of Patients during this outbreak. For instance, some hospitals started to set up the system of screening the ER patients by their disease of communicable and non-communicable type. Plus, private organizations such as media, transportation, many business associations had done their job to raise awareness of the importance of hygiene campaigns like proper and frequent handwashing and coughing on sleeves. These voluntary movements also could be an element of dealing with 2015 MERS-CoV in Korea. Second, some individual- and hospital-level information was not adjusted for in this study because of data limitation. The transmission of MERS-CoV in Korea occurred mainly in hospitals [16]. We assumed it at the similar environment. People also avoided outings as a preventive measure against MERS-CoV. Therefore, we could not adjust for these factors. Further research including more individual and hospital level data is needed. Third, it took 2–3 days to confirm MERS-CoV infection via PCR test. The time taken to record PCR test results might be dependent on laboratory capacity. However, we conducted sensitivity analyses and our results were consistent.

## 5. Conclusions

In summary, we found that providing timely and accurate information to the public during an outbreak was crucial to both risk communication and the control and prevention of further infection. Therefore, health authorities should bolster their capacity to disclose information in a timely, accurate, and transparent way.

## Figures and Tables

**Figure 1 ijerph-17-00305-f001:**
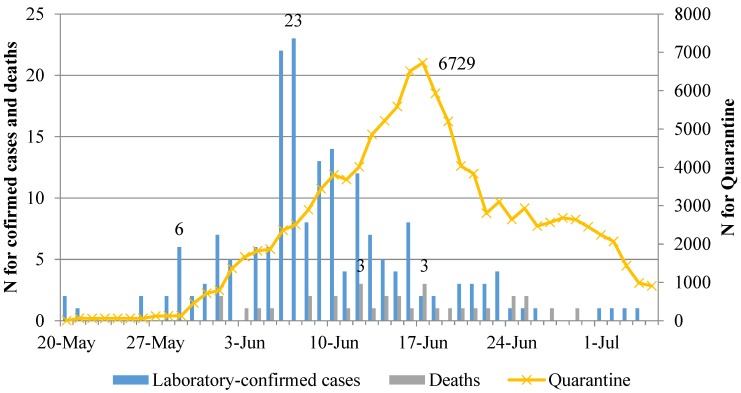
Daily statistics for laboratory-confirmed Middle East respiratory syndrome coronavirus (MERS-CoV) cases, deaths, and quarantine.

**Figure 2 ijerph-17-00305-f002:**
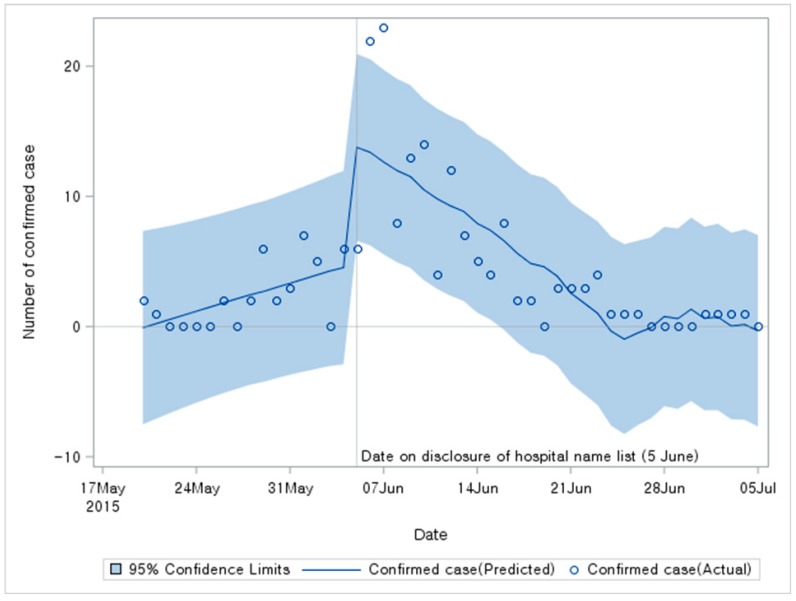
Results of segmented regression for confirmed cases.

**Table 1 ijerph-17-00305-t001:** Effects of disclosure of hospital list.

	Laboratory-Confirmed Cases		
	Estimate	Standard Error	*p*-Value
Intercept	−0.406	1.743	0.817
Time	0.316	0.180	0.087
Disclosure of the hospital name list	9.632	2.057	<0.001
Time after Disclosure	−0.699	0.198	0.001
Quarantine_t-8_	−0.001	0.0004	0.012
R-square	0.64		
DW	2.13 ^†^		

^†^*p* > 0.05.

**Table 2 ijerph-17-00305-t002:** Results of sensitivity analysis of the time taken to complete a PCR test.

	Time of PCR Test			
	0 Day		2 Days	
	Estimate	*p*-Value	Estimate	*p*-Value
Intercept	−0.166	0.932	−0.543	0.731
Time	0.274	0.205	0.339	0.032
Disclosure of the hospital name list	9.785	<0.001	9.463	<0.001
Time after Disclosure	−0.658	0.006	−0.727	<0.001
Quarantine_t-7+time of PCR test_	−0.001	0.062	−0.001	0.004
R-square	0.59		0.69	
DW	1.69 ^†^		1.67 ^†^	

^† ^*p* < 0.05.

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
