# Peer review of "Effect of Information Disclosure Policy on Control of Infectious Disease: MERS-CoV Outbreak in South Korea"

_ijerph, 2020, doi:10.3390/ijerph17010305_

Round 1

Reviewer 1 Report

An interesting manuscript showing a significant reduction in the outbreak of MERS-V in South Korea after public dissemination of affect hospitals.  Methods appear to be adequate.  Only criticism is that other factors that may have influenced the results are not discussed to any extent.  Example includes the role of the MERS management task force and other government agencies.  Their actions during this outbreak should be discussed as possible explanations for the reduction in the number of cases.

Author Response

We appreciate your comments. We discussed the other factors including health authority, hospitals and the private sector in the discussion section as below;

“First, the information dissemination was likely to lower the possibility of the public’s further contracting MERS-CoV, as it helped members of the public to avoid exposed hospitals with the given information by the health authority [13]. Following a news article, hospital utilization had decreased by up to 50% since May 20, 2015. This might have occurred because citizens were reluctant to visit hospitals as they were afraid of contracting the virus [2]. Therefore, the way citizens had stayed vigilant not to catch the virus was likely to be strengthened after the disclosure of the hospital names. Second, since the citizens had received the information regarding the affected hospitals, people who had visited one of those hospitals or contact friends and family who went to the affected medical facilities would be more likely to report to the KCDC when they experienced MERS symptoms such as fever, vomit, sore throat, vomiting, diarrhea and follow the prevention and control guidance voluntarily and effectively [14]. These mechanisms are supposed to have contributed to the reduction in the number of confirmed cases.”

Although analysis using additional data was difficult due to the data limitations of this study, we added future research suggestions in the limitation section as below;

“First, we did not analyze the effects of other noteworthy variables such as the increased number of health authorities’ staff, medical facilities’ capacity bolstering, the strengthened engagement of media and other private sectors over time. For example, increases in the number of staff members in the Central MERS Management Task Force played an important role in preventing and controlling the outbreak. Therefore, the size of the MERS Rapid Response Team dispatched to exposed hospitals should be included in further studies. Besides, hospitals regardless of having MERS patients at that time in Korea had been more involved in detecting and controlling outpatients and visitors as time went by. Therefore, this factor could be one of the reasons to help decrease the number of Patients during this outbreak. For instance, some hospitals started to set up the system of screening the ER patients by their disease of communicable and non-communicable type. Plus, private organizations such as media, transportation, many business associations had done their job to raise awareness of the importance of hygiene campaigns like proper and frequent handwashing and coughing on sleeves. these voluntary movements also could be an element of dealing with 2015 MERS-CoV in Korea. Second, some individual- and hospital-level information was not adjusted for in this study because of data limitation. The transmission of MERS-CoV in Korea occurred mainly in hospitals [16]. We assumed it at the similar environment . Also, people avoided outings as a preventive measure against MERS-CoV. Therefore, we could not adjust for these factors. Further research including more individual and hospital level data is needed.”

Reviewer 2 Report

In this study, “Effect of information disclosure policy on control of infectious disease: MERS-CoV outbreak in South Korea”, Noh et al. examined the effect of disclosing the hospital information with MERS-CoV patients during the outbreak in South Korea from May 20, 2015 to July 5, 2015. Using segmented linear regression, they found that there is a significant decrease in laboratory confirmed cases following disclosure of a list of hospitals with MERS-COV cases. The sensitivity analysis results supports this finding. Overall, the manuscript is written clearly and I do not have major corrections. 

Minor corrections:

Discussion, para 2, line 2: I believe “MERS-Co” is a typo. Discussion para 3, line 3-4: Please check for grammar. (“despite there was criticism...”) Discussion, page 8, line 1-2: It is unclear what this sentence means. 

Author Response

We appreciate your comments. We revised all minor corrections in the discussion section and revised the sentence in line 1-2 clear as below;

“Following the outbreak of SARS in Singapore, most people emphasized the importance of current study results and the disclosure of crucial information to the public and the media, as to what would have helped during the crises.”

Round 2

Reviewer 1 Report

All concerns from the review were adequately addressed in the revised submission